# Multifaceted Roles of ALK Family Receptors and Augmentor Ligands in Health and Disease: A Comprehensive Review

**DOI:** 10.3390/biom13101490

**Published:** 2023-10-07

**Authors:** Luka Katic, Anamarija Priscan

**Affiliations:** 1Department of Medicine, Icahn School of Medicine at Mount Sinai Morningside/West, 1000 Tenth Avenue, New York, NY 10019, USA; 2Department of Pharmacology, Yale University School of Medicine, New Haven, CT 06520, USA; 3Department of Immunobiology, Yale University School of Medicine, New Haven, CT 06520, USA; anamarija.priscan@yale.edu

**Keywords:** receptor tyrosine kinases, ALK, Augmentor α, Augmentor β, LTK, signaling pathways, central nervous system biomolecules, pain modulation pathways, therapeutic development, ALK inhibitor side effects

## Abstract

This review commemorates the 10-year anniversary of the discovery of physiological ligands Augα (Augmentor α; ALKAL2; Fam150b) and Augβ (Augmentor β; ALKAL1; Fam150a) for anaplastic lymphoma kinase (ALK) and leukocyte tyrosine kinase (LTK), previously considered orphan receptors. This manuscript provides an in-depth review of the biophysical and cellular properties of ALK family receptors and their roles in cancer, metabolism, pain, ophthalmology, pigmentation, central nervous system (CNS) function, and reproduction. ALK and LTK receptors are implicated in the development of numerous cancers, and targeted inhibition of their signaling pathways can offer therapeutic benefits. Additionally, ALK family receptors are involved in regulating body weight and metabolism, modulating pain signaling, and contributing to eye development and pigmentation. In the CNS, these receptors play a role in synapse modulation, neurogenesis, and various psychiatric pathologies. Lastly, ALK expression is linked to reproductive functions, with potential implications for patients undergoing ALK inhibitor therapy. Further research is needed to better understand the complex interactions of ALK family receptors and Aug ligands and to repurpose targeted therapy for a wide range of human diseases.

## 1. Introduction

This review marks the 10-year anniversary of the discovery of physiological ligands Augα (Augmentor α; ALKAL2; Fam150b) and Augβ (Augmentor β; ALKAL1; Fam150a) for anaplastic lymphoma kinase (ALK) and leukocyte tyrosine kinase (LTK), which were previously considered orphan receptors [1]. While small-molecule ALK inhibitors that target the intracytoplasmic kinase domain by binding to the ATP-binding site have been extensively studied and used for cancer treatment, their repurposing for other pathologies, such as neurological, cardiac, and metabolic diseases, has proven challenging [2,3,4]. Given their need to be lipophilic to cross the cell lipid bilayer and bind intracellularly, this class of drugs is prone to toxicities [5]. Moreover, the structural similarities between the ATP binding sites of kinase domains in the entire family of receptor tyrosine kinases (RTKs) make these drugs relatively non-specific, increasing the possibility of off-target binding and side effects [5].

Monoclonal antibodies that target ALK receptors offer an alternative approach, as they are highly specific, generally non-toxic, and can fine-tune signaling. By blocking ligand binding or promoting dimerization, monoclonal antibodies have the potential to modulate ALK signaling pathways for therapeutic benefit in non-cancerous pathologies [6,7,8,9]. They generally do not cross the lipid bilayer or the blood-brain barrier [10]. Activating analogs of a ligand can be used as an agonist to bind to the ligand binding site [11]. This review will focus on the physiological relevance of ALK-Augα and LTK-Augβ signaling and encourage further research efforts toward the repurposing of monoclonal antibodies, Aug analogs, and TKIs to modulate these mechanisms of action. 

## 2. Biophysical and Cellular Background

The receptor tyrosine kinase field encompasses 20 families, comprising approximately 55 known receptors to date [12]. Its beginnings can be traced back to the 1970s [13], and since then, significant advancements have been made. These include gaining biophysical structural understanding, identifying receptor functions (such as proliferation, differentiation, survival, and metabolism), and developing a wide range of therapeutic agents that modulate RTK functions, many of which have been approved by the FDA or are currently in clinical trials [12].

A typical receptor features an extracellular domain, a single transmembrane helix, and an intracellular domain. When a growth factor binds to the extracellular domain, it activates signaling through the induction of dimerization. The self-association of two receptors then propagates to the intracellular part, where the tyrosine kinase domains phosphorylate one another. In all activated RTKs, the activation loop within the N lobe of the TKD undergoes a conformational shift from a cis-autoinhibited state to a trans-phosphorylation-prone state, ultimately leading to the opening of the ATP binding sites within the kinase domain, which, in turn, induces trans-phosphorylation and activation [12].

Over the past several decades, extensive research has been conducted on receptor tyrosine kinases (RTKs), leading to the identification of most RTK ligands. However, the ALK family remained an exception [1,12]. This family consists of only two receptors: ALK and LTK [14,15]. These receptors share striking similarities in their structure, including an intracellular kinase domain, a transmembrane helix, and an extracellular part. In the extracellular region, the C-terminal portion is conserved between the two receptors, featuring an EGF-like domain, a glycine-rich region, and a TNF-like domain. Additionally, ALK possesses a distinctive N-terminal segment comprising two MAM domains, an LDLa-like region between them, and an N-terminal heparin-binding region [16,17,18,19,20].

In lower evolutionary species, the ALK family has only one receptor, ALK [19]. The LTK receptor likely originated from gene duplication in vertebrates, with lower evolutionary vertebrates possessing the unique N-terminal part [19]. However, in mammals, the N-terminal portion of LTK is absent. The ALK phylogenetic tree reveals that lower evolutionary species such as *D. melanogaster* [21] and *C. elegans* [22] maintain the N-terminal part, resembling the human structure, while some vertebrates, like *D. rerio*, have lost portions of the N-terminal region of the ALK receptor [23]. Intriguingly, the N-terminal part of human ALK is proteolytically processed in the brain but not in dorsal root ganglia (DRG) neurons [24]. Matrix metalloproteinase 9 (MMP9) mediates this proteolytic processing (shedding) of ALK’s N-terminal region [25].

The initial ALK ligands identified were for *D. melanogaster*’s ALK (dALK) and *C. elegans*’s ALK (SCD-2). The ligand for dALK is Jeb (Jelly Belly), while the ligand for SCD-2 is Hen-1 (Hesitation Behavior 1). Interestingly, both ligands possess an LDLa-like domain. Hen-1 is associated with mediating learning, whereas Jeb is involved in promoting visceral muscle fusion. Nevertheless, neither ligand binds to the vertebrate ALK family, and no vertebrate homologs have been discovered [26,27,28] (Figure 1).

The first suggested human ALK ligand was long-chain heparin (as well as other glycosaminoglycans). This was because heparin triggered ALK phosphorylation in neuroblastoma (NB) cells, but it failed to activate ALK in 3T3 cells. These findings implied that another unidentified factor might be necessary, in conjunction with heparin, to induce ALK activation [16].

In 2014, a screening of 3191 extracellularly secreted proteins revealed that Fam150b (Augα) and Fam150a (Augβ) function as ligands for the LTK receptor. LTK phosphorylation increased more than 30-fold with Augβ and over two-fold with Augα compared with the control. Interestingly, IGF-2 also elevated LTK phosphorylation above baseline but with significantly weaker potency [1]. No homologs of Augs have been found in non-vertebrate species, and no homologs of Jeb or Hen-1 have been identified in vertebrate species [29].

The discovery of Augs in the LTK screen was later applied to ALK. Both Augs were found to induce ALK phosphorylation when co-expressed in cells, an effect that was reversible upon ALK inhibition using the tyrosine kinase inhibitor (TKI) crizotinib and the ALK-specific monoclonal antibody mAb13. Additionally, co-expression of Augs and ALK in Drosophila eye produced an ALK gain-of-function rough eye phenotype, demonstrating in vivo relevance [30].

An examination of the hierarchy and specificity of ALK family receptors and Aug ligands revealed that Augα activated both ALK and LTK at picomolar concentrations, while Augβ only activated LTK at picomolar concentrations and ALK at nanomolar concentrations. The authors also demonstrated that the unique N-terminal part of ALK was not crucial for Aug-induced stimulation, and heparin exhibited synergy only with lower concentrations of Augα [17].

Researchers reported issues with Aug stability, as, being small and basic proteins, they are susceptible to proteolytic cleavage [17,30]. Furthermore, full-size Augα ligands tended to precipitate at concentrations higher than 0.3 mg/mL [18,31]. To address this problem, scientists employed various approaches, ultimately leading to the discovery of a biologically active fragment of Aug. Full-size Augα and Augβ share two distinct regions: an N-terminal variable region (VR) that differs between Augα and Augβ and a C-terminal Aug domain (AD) that is conserved between the two ligands. Four cysteines in the AD region form two intramolecular disulfide bridges. Full-size Augα is expressed as a disulfide dimer, with cysteines at position Cys66 joined together, while full-size Augβ, which lacks an additional cysteine, is expressed as a monomer. Isolating the AD reduced dimer properties, but both truncated and full-size versions of Augα exhibited the same phosphorylation properties, neurite outgrowth properties, and kinetics specific to ALK/LTK [31]. It is currently uncertain whether the variable region (VR) of Augs undergoes physiological processing or if it influences different functions depending on its location or expression. 

The three reported ALK family receptor and augmentor structures have been studied by the Kalodimos, Schlessinger, Klein, and Savvides research groups (Table 1). The first two groups propose a 2:2 complex (two receptors with two ligands), while Savvides suggests a 2:1 complex (two receptors with one ligand) [18,19,20]. 

Savvides’ group first published findings on the isolation of human ALK and LTK ligand-binding segments, and their biophysical analysis using Augβ and Augα fragments. Their results indicate that the 2:1 assembly could be a possible mechanism for LTK-Augβ dimerization and activation. Given the size exclusion chromatography results of the ALK-Augα complex within the study, the possibility that crystallographic conditions may have influenced the results cannot be excluded [19].

A second publication from Kalodimos and Schlessinger’s groups focused on ALK and used full-size Augα ligands, overcoming stability and precipitation issues by mutating Cys66 into hydrophobic tyrosine. Their findings support a 2:2 complex model with four interaction sites. Although the membrane is not incorporated in the complex, the authors propose an additional fourth interaction site where Augα binds to the membrane because of its orientation relative to the EGF-like domain, which points toward the transmembrane segment. The positively charged residues of Augα (Lys95, Lys96, Lys99, His100) could interact with the negatively charged membrane. When these residues are mutated, no ALK phosphorylation is detected in cells. The complex’s structure and interactions suggest that positively charged Augα residues and EGF-like domain bending are crucial for ALK activation [18].

Lastly, a third manuscript by Klein and Schlessinger’s groups used a truncated ALK connected to the Augα AD connected to a linker, resulting in a 2:2 receptor–ligand structure with three interaction sites. Their results indicate that site mutations can impact ALK phosphorylation properties in cells and that certain antibodies can inhibit or activate ALK by binding to specific sites [20].

Research on ALK family receptors and augmentors has produced varying models of their structures and complexes (Figure 2). While some groups propose a 2:2 complex, others suggest a 2:1 complex. Further studies are needed to clarify the discrepancies and better understand the mechanisms of dimerization and activation in these receptors. Furthermore, the function, role, and biophysical properties of ALK’s distinctive N-terminal segment have yet to be fully elucidated.

## 3. Cancer

The development of cancer through ALK receptor signaling can be initiated by either an ALK fusion protein or a point mutation in the full-length ALK receptor [32]. Notably, the oncogenic activity of the ALK fusion protein, characterized by receptor dimerization and activation, is driven solely by its oligomerization and is entirely independent of any involvement with the ligand Augα [32,33]. EML4 is a common ALK fusion protein that physiologically oligomerizes inside the cell, creating machinery for ALK dimerization and activation. This inherent machinery keeps the protein within the cell, so it is physically impossible for Augα to exert any influence on the activation of the ALK fusion protein [34] A variety of fusion ALK proteins have been reported in numerous cancers, including NSCLC, IMT, and ALCL, with fusions developing almost exclusively on the common breakpoint in exon 20 [32,35]. The ALK locus is particularly prone to chromosomal fusions, and this appears to play a role in both cancer development and evolution [19,35]. TKIs are effective against ALK fusion protein-driven cancers, but resistance and relapse usually occur after initial remission [36,37]. It is often difficult to determine if the receptor mutation is solely causing inhibitor resistance or if it is also contributing to oncogenic activity, as is the case with the ALK-L1198F and ALK-G1201E mutations. They were considered oncogenic, but after the discovery of Augs, these "oncogenic mutants" were tested for a ligand response, and it was found that they do not result in ligand Augα-independent ALK activation [38,39].

Almost exclusively found in neuroblastoma, point mutations have been observed in the full-size ALK receptor, but no fusion proteins have been reported in this type of cancer [32,35]. About 6–12% of all sporadic neuroblastomas are ALK-mutation-positive, with ALK neuroblastoma mutants having three subtypes: ligand-independent, ligand-dependent, or kinase-dead mutants that contain mutations of downstream signaling proteins such as the RAS/MAPK pathway, which includes NF1 mutations and RAF mutations [32]. Although neuroblastoma is strongly associated with altered ALK signaling, ALK gain-of-function mice develop sympathetic ganglia hyperplasia and not cancer, and additional key triggers are necessary for cancer development, typically MYC amplification [40,41,42,43,44]. MYC transcriptional factor drives ALK expression, and ALK signaling drives MYC expression, creating a positive feedback loop [45,46]. MYC-driven neuroblastoma mouse models typically exhibit incomplete penetrance and late onset, as only around 50% of the mice develop cancer within an average of 40 weeks. However, when MYC-driven neuroblastoma mouse models were crossbred with mice that have an ALK gain-of-function mutation (ALK-F1178S), all the mice (98%) developed neuroblastoma within an average of 28 weeks [47]. Clinical trials of neuroblastoma ALK-mutation-positive subgroups treated with crizotinib were not successful, but new, more potent ALK TKIs are now being tested [48]. The discovery of Augs has led scientists to expand the patient subpopulation that could be targeted with ALK inhibitors, as Augα overexpression in mice almost completely reproduces ALK-MYC synergy in driving neuroblastoma [47]. The same synergy is found between MYC-driven neuroblastoma and Augα overexpression, suggesting that inhibitory antibodies might be effective for patient subgroups with ligand misregulation in addition to the potential benefits of ALK TKI treatment [47].

MYC, ALK, and Augα are all located on the 2p chromosome, and the exact 2p gain region is associated with the development of neuroblastoma [49,50]. ALK TKI treatment could be beneficial for all NB patient subtypes with the 2p gain region. An analysis of 356 NB cancer samples showed that 107 patients (30%) had 2p gain (more than four copies) or amplification (more than eight copies), most of which contained Augα, MYCN, and ALK genes [49]. The number of somatic mutations in a significant proportion of NB cases is typically low, and there are no significant hotspot gene mutations. Rather than being driven by mutations, NB seems to be a cancer that is primarily determined by gene copy number [51]. Interestingly, ALK mutation and amplification are prevalent mostly when MYCN is amplified, but Augα amplification is associated with non-MYCN-amplified cases [49]. The Augα/ALK/MYCN signaling axis could be the key relationship between 2p gain and oncogenic activity. Additional evidence supporting this theory is a case report of a 6-month-old boy with poorly differentiated unfavorable NB. After 10 months of chemotherapy, his cancer developed 2p gain and TrkA overexpression [52]. In general, high TrkA expression is a marker for favorable prognosis since it could lead to spontaneous regression, but TrkB expression is a marker of poor prognosis [53]. This patient was started on entrectinib (ALK and Trk inhibitor), improved significantly, and maintained a stable clinical situation more than 5 years after clinical diagnosis [52]. In addition to this case, multiple cases of germline 2p gaining partial trisomy have been reported to trigger NB development [54,55,56].

In addition to 2p gain and MYC amplification, 11q deletion is another significant trigger for NB development [50]. This deletion is believed to remove the tumor suppressor genes DLG2 and SHANK2, which drive neuroblastoma differentiation from proliferation [57,58]. Interestingly, Augα/ALK/MAPK signaling has been shown to both suppress DLG2 transcription and induce NB neurites and differentiation [31,58]. NB can be further classified based on its level of differentiation, with a more benign state having more differentiated adrenergic cells, while a more malignant state resembles neural crest cells and mesenchymal identity [59]. The RTKs TrkA and RET are major drivers of differentiation in these cells [60]. According to Siaw et al. [61], RET expression is high in more differentiated neuroblastoma (NB) cells, while using CRISPR to knock out RET leads to a strong transition toward an undifferentiated mesenchymal state. ALK-positive NB cells activate and phosphorylate RET within minutes of being exposed to Augα, but the RET ligand GDNF does not activate ALK. This activation is reversed by inhibiting ALK or MAPK [61,62]. Lambertz et al. [63], on the other hand, reported that RET signaling occurs through PI3K/AKT/mTOR, but in either case, RET signaling is downstream of ALK [61]. Targeted therapy with joint ALK and RET inhibitors should be used with caution given the complexity of differentiation.

According to a recent study, Augβ was linked to the progression, migration, and invasion of colorectal cancer, as it regulates the Sonic Hedgehog (SHH) signaling pathway that drives tissue polarity and invasion [64]. However, another study failed to detect any Augβ expression in colon cancer cell lines but instead found high levels of both ALK and Augα in cell lines similar to the Consensus Molecular Subtype 1 (CMS1) of colon cancer [65]. This subtype is characterized by DNA mismatch–repair-system-induced mutations, and its growth and proliferation depend on ALK-Augα signaling through the AKT pathway, which can be reversed by Crizotinib (ALK TKI) [65,66]. Interestingly, Crizotinib induces proliferation in cell lines exhibiting CMS-4, suggesting that ALK TKI is not universally beneficial for colon cancer patients [65]. The cancer-reducing effect of ALK TKI on CMS1 was demonstrated in a 3D spheroid model and in a physiologic mouse model, supporting the potential for a clinical trial using combination therapy containing ALK TKI to treat the CMS1 subtype of colon cancer [64,65,66].

## 4. Metabolism and Body Weight Regulation

The first report of ALK kinase domain deletion in mice demonstrating differences in body weight was published during an investigation of the endogenous role of ALK in the CNS [67]. This study was followed up by GWAS data mining on the healthy Estonian population (Estonian Biobank), which identified a thinness-associated variant located in the first intron of ALK [68]. Knockdown of ALK caused by RNAi in Drosophila reduced triglyceride levels without affecting food intake. ALK knockout mice also displayed a thin phenotype, which was due to sympathetic overactivity and a reduction in fat tissue mass, while lean mass tissue remained intact. To clarify the origin of this thin phenotype, floxing genetic models were used, and ALK was deleted using Cre recombinase from specific tissues: white adipose tissue (AdipoqCre), brown adipose tissue (Ucp1Cre), intestine (VillinCre), liver (AlbuminCre), muscle (MckCre), and hematopoietic cells (Vav1-iCre). However, all of the genetic models failed, but a stereotactic injection of AAV:GFP viral vector containing Cre recombinase into the paraventricular nucleus (PVN) area of a genetically modified mouse containing LoxP sites on the 3’ and 5’ end of the ALK gene displayed a thin phenotype. In this region, neurons containing ALK were preferentially excitatory glutamatergic neurons of pneSS and pneCRH. Other synaptically interconnected neurons that were not mapped were also infected with the virus with subsequent ALK deletion [68]. To further expand the mechanism of signaling, the upstream deletion of potential ALK ligands was generated. Augα knockout mice and Augα/Augβ double-knockout mice displayed a thin phenotype and sympathetic overactivity with only fat tissue mass reduction, while lean muscle mass was preserved. It was proposed that this observed thin phenotype originates from Augα in Agouti-related peptide (AgRP) neurons, which display projections toward PVN. Augα is also upregulated following starvation in AgRP neurons. Other neurons expressing Augα were the paraventricular nucleus (PVN), the dorsomedial nucleus (DMH), and the suprachiasmatic nucleus (SCN). The authors also showed that Augα and ALK signaling are responsible for the pathway and projection development between AgRP neurons and PVN neurons [69].

The regulation of sympathetic activity is also influenced by afferent parasympathetic vagal neurons, which are crucial for transmitting visceral pain and mediating vagal reflexes [70]. Nucleus tractus solitarius (NTS) acts as the second-order neuron for n.vagus afferent projections and has a well-established axonal connection to PVN [71]. There is a strong possibility that the AAV:GFP viral vector used by Orthofer et al. to infect PVN may have also infected NTS axonal projections. Given that Augα is highly expressed in DRG and plays a role in pain signaling [72], investigating its expression in other peripheral ganglia within the autonomic nervous system may shed light on the mechanisms of body weight regulation.

However, inhibiting ALK-Aug signaling to treat obesity may pose a challenge, as small molecular inhibitors of ALK that penetrate the blood–brain barrier can lead to weight gain, hypertriglyceridemia, hypercholesterolemia, and hypertension [73]. For instance, Lorlatinib has been linked to weight gain in children being treated for neuroblastoma [74], and a transcriptome-wide association study has linked the Augα gene to childhood obesity [75]. Furthermore, another brain-penetrating ALK inhibitor, repotrectinib, has also been associated with weight gain [76]. These findings suggest that multiple excitatory and inhibitory neuron groups containing Alk and/or Augα may be involved in weight regulation, adding complexity to the current data evidence.

## 5. Pain Signaling

Pain is a multifaceted experience involving different types of pain stimuli and nociceptor receptors [77,78]. Based on the etiology, pain sensation can be classified into three subgroups: mechanical, thermal, and chemical pain. Mechanical pain arises from stimuli impacting free nerve endings, TRPA1 receptors, and other similar structures, while thermal pain results from the activation of TRPV1 and TRPM8 receptors and related receptors because of temperature changes. Chemical pain, on the other hand, is triggered by substances that interact with receptors such as TRPV1 and ASIC [78]. The sensitivity and signaling of these nociceptor receptors can be modulated either from the outside of a neuron, for example, during inflammation, or from within a neuron (neuropathy) caused by injury, diabetes, viruses, etc. [78,79] Chronic pain resulting from these modulations is a significant problem for patients, as current treatment options targeting the molecular etiology of the problem are limited [78]. Therefore, an ideal pain treatment should not alter the baseline pain sensation [80]. Behavioral tests have been developed to approximate the pain sensation experienced by mice, including the Von Frey filament assay for mechanical pain and the Hargreaves assay for thermal pain [81].

MAP kinase is a potential molecular target for modulating pathologic pain states, as it highly regulates the ionic channels that initiate and transmit pain, likely through indirect phosphorylation and/or changes in nociceptor transcription and expression [82,83]. Since MAP kinase can be activated by the entire family of receptor tyrosine kinases [12], researchers should focus on spatiotemporal regulation by the MAP kinase and its upstream effector receptor. Augα-ALK interactions drive the MAP kinase pathway and modulate pain signaling. According to the currently proposed mechanism, Augα from the dorsal root ganglia (DRG) stimulates ALK in the spinal cord’s dorsal horn, resulting in pain signaling modulation. Augα is highly expressed in DRGs and is upregulated two-fold following inflammation. Injecting Augα into the spinal cord’s cerebrospinal space stimulates ALK phosphorylation in the dorsal horn, making mice more sensitive to thermal pain. Inhibiting the ALK kinase domain with the blood–brain-barrier-penetrating ALK inhibitor Lorlatinib and depleting Augα with an antisense oligodeoxynucleotide intrathecal injection can prevent inflammation-induced and neuron-injury-induced increased pain sensitization [72]. However, it is unclear if Augα from DRGs creates signals in the synaptic space of the dorsal horn, inducing ALK phosphorylation. Augα is also expressed in the spinal cord, and DRG-primary cultures show signs of having the Augα receptor ALK and/or LTK [24,72]. Augα modulates the DRGs’ ionic channel regulation and its electrophysiological properties, inducing an increased action potential frequency and neurite outgrowth in DRG neurons in response to stimulation [72]. It is possible that two distinct signaling circuits of Augα and ALK signaling exist, one in the DRGs and one in the spinal cord. DRGs are outside the blood–brain barrier, making them an easily targeted location for antibody modulation [84].

The inability of adult neurons to proliferate [85] suggests that the overexpression or mutation of receptor tyrosine kinases driving MAPK signaling may contribute to the development of neuropathic pain [82,83]. A group of isolated cancer patients with neuropathic pain showed a significant response to EGFR inhibitors [86,87], while a meta-analysis of ALK inhibitors intended to treat NSCLC revealed that a subgroup of patients experienced reduced musculoskeletal pain [88]. Anti-NGF monoclonal antibodies (TrkA receptor inhibitors) have shown promise in treating pain and are currently in clinical trials, but a small subset of patients develop rapidly progressive joint degeneration [89]. RET, FGFR1, KIT, InsR, and PDGFRβ are among the other receptors associated with pain modulation within DRGs [90,91,92,93,94], and the understanding of the subtypes of neurons expressing each of these receptors and the spatiotemporal relationship between them driving MAPK in DRGs remains to be elucidated. CRISPR has shown promise as a technique for effectively studying and modulating the epigenome of primary cultured DRGs with a 75% success rate [95].

In addition to nociceptors, DRGs contain neurons with other somatosensory functions, and LTK KO mice show impaired coordination and balance on the rotarod test [96], while dizziness is one of the major symptoms of ALK inhibitors [88]. Augα expression in DRGs [72] suggests that proprioception signaling may also be regulated by Augα and LTK.

## 6. Ophthalmology and Pigmentation

Numerous clinical cases of patients with ALK-driven cancers who were treated with ALK inhibitors have reported visual impairment as a common side effect, with symptoms such as blurred vision, lowered visual acuity, presbyopia, flashes, and accommodation disorders being strongly correlated with ALK inhibitors [88,97]. In mice, ALK expression has been found in various parts of the eye, including the ciliary ganglia, the neural and pigmental layers of the retina, the lens, and the corneal epithelium [98]. Augα expression has been detected in mice’s nociceptive neurons situated in the dorsal root ganglia (DRG) when they innervate the trunk. It is likely that Augα is also expressed in nociceptive neurons within the trigeminal ganglia, where they are responsible for innervating the eye [72,99]. However, the expression of Augα, Augβ, ALK, and LTK in other autonomic ganglia (sympathetic or parasympathetic), in the context of eye innervation, remains unknown. In Drosophila, ALK and its ligand Jelly Belly play an important role in the development of the visual pathway, as they stimulate projections between the optic lobe from the visual cortex and photoreceptor R8, which are crucial for color vision [100]. This finding has been used to study ALK signaling in vivo, as the ectopic expression of mutated human ALK (F1174S), which has a gain of function, leads to a strong rough eye phenotype [101].

In zebrafish, the homologs of human Augα and Augβ are DrAugαa, DrAugαb, and DrAugβ, which signal through the zebrafish homolog of human LTK, DrLTK, to stimulate eye pigmentation. DrAugαa is critical for truncal-iridophore-induced pigmentation, while DrAugβ is necessary for operculum’s iridophore-induced pigmentation, and both are essential for eye-iridophore-induced pigmentation. At the larval stage, DrAugαa and DrAugαb induce eye iridophore pigmentation, and the interaction of all three zebrafish Augs seems to play a significant role in the eye [23,102]. Interestingly, LTK gain-of-function mutation stimulates iridophore growth and proliferation rather than migration and localization [23,103]. DrLTK has similar extracellular domains as human ALK but shares intracellular kinase domain homology with hLTK. Additional studies are required to clarify the receptor properties of zebrafish LTK and to translate this phenotype into higher evolutionary organisms such as mice.

Neural-crest-derived cells, iridophores, play a crucial role in the coloration of fish at both the larval and adult stages, and LTK, Augα, and Augβ are strongly expressed in these cells [23]. Larval iridophores are directly derived from neural crest cells [104], while adult iridophores arise from stem cells in the dorsal root ganglia [105]. Humans lack iridophores, but melanocytes that induce coloration arise from the neural crest [106]. Recently, LTK expression was linked to the melanoid color variant of amphibian axolotls [107]. In an animal model of uveal melanoma, ALK was found to be positive in all samples, and crizotinib prevented their metastasis [108,109]. Epithelial-cell-type uveal melanomas in humans are more frequently ALK-positive than benign spindle-cell-type cases [108]. However, the expression of LTK and Augs in melanomas is currently unknown. It is necessary to determine the precise role of ALK, LTK, and Augs in neural crest development.

## 7. CNS and Teratogenicity

For a long time, it has been known that ALK and LTK are expressed in the central nervous system (CNS) [110] ALK is known to play critical roles in modulating the architecture and strength of synapses [111]. Studies on knockout (KO) mice have shown that ALK deficiency leads to a 50% decrease in neurogenesis, whereas LTK deficiency does not have any effect on neurogenesis. Interestingly, ALK-LTK double-KO mice have an 80% decrease in neurogenesis, suggesting that LTK may partially compensate for ALK deficiency. Even though synaptogenesis and neurogenesis are impaired, brain volume remains normal [96]. It is worth noting that, although neurogenesis decreases in these mice, it does not result in any devastating phenotype, as has been observed in viable mice [67,96,112,113].

In contrast, a CNS-penetrable ALK inhibitor, Lorlatinib, has shown severe teratogenicity in mice, leading to malformations and abortion (Pfizer report). However, in humans, two clinical cases of pregnant patients treated with alectinib resulted in normal childbirth and development [114,115]. During embryological development, ALK is present throughout the CNS, including in specific regions of the brain such as the thalamus, mid-brain, olfactory bulb, and ganglia [98,110]. The expression of ALK, together with LTK, in the hippocampus and amygdala is associated with memory and anxiety. Interestingly, ALK KO mice show enhanced cognitive performance and reduced anxiety, while LTK KO mice do not show any significant changes [96]. In Drosophila, ALK inhibition has been shown to enhance olfactory-connected learning [116].

However, ALK KO mice consume greater doses of alcohol, and the mechanism behind this phenotype appears to be within the nucleus accumbens, where ALK regulates excitatory synapses from post-synaptic neurons. The loss of ALK enhances the excitatory transmission in nucleus accumbens neurons [113]. ALK also regulates the behavioral response to cocaine [112]. ALK has been associated with other psychiatric pathologies, including schizophrenia, but additional research is needed [117]. Hopefully, further research on ALK-Augα and LTK-Augβ signaling pathways will help identify the precise roles of these proteins in various neurological and psychiatric pathologies.

## 8. Reproductive System

The expression of ALK in both the testes and ovaries is well established in the scientific literature [98,110]. Young male patients who are prescribed ALK inhibitors such as crizotinib and ceritinib may experience endocrine side effects, which can lead to symptoms resembling hypogonadism and severely reduced testosterone levels. Studies have shown that this is likely due to CNS-mediated mechanisms, as FSH and LH levels in these patients are low. Fortunately, these side effects are reversible upon drug discontinuation and can be treated with testosterone supplements [118,119]. Interestingly, mice with ALK knockouts (KOs), either from exon 1 deletion or kinase deletion, have exhibited delayed breeding, which seems to be associated with puberty delay. Prepubertal ALK KO mice also show mild disorganization in seminiferous tubules. Testosterone levels, hypothalamic GnRH neuronal levels, and Sertoli cell numbers are low at the prepubertal stage, but they normalize once mice enter puberty [119]. Puberty delay might be beneficial for patients with precox puberty, which is becoming a growing concern [120]. Additionally, treating WT mice with crizotinib (at a high 20 mg/kg concentration) has been shown to decrease adult testosterone levels [119]. Whether this observation is physiologic remains to be elucidated with additional research efforts involving the role of Augs in reproduction.

## 9. Conclusions and Perspectives

As demonstrated by various studies, the signaling pathways involving ALK-Augα and LTK-Augβ play a significant role in several biological systems (Figure 3). Thus, it is crucial for clinicians prescribing ALK inhibitors to consider the roles that these pathways play in normal human physiology. Additionally, understanding the mechanisms of action through structural and biophysical studies has helped in the development of monoclonal antibodies that may be repurposed to treat specific types of cancer and pathological pain [20]. Brain-penetrating TKIs also hold potential for modifying psychiatric and neurological disorders [96], metabolic syndrome [69], and precocious puberty [119]. Moreover, based on the ophthalmological phenotypes described, analogs of the AD could potentially be used as eye drops [23,102]. When key mechanistic insights are discovered, therapeutic development becomes trivial. Therefore, it is essential for the academic community to continue its efforts in clarifying the roles of Augs in the mentioned phenotypes and potentially uncovering new physiological roles for these signaling cascades. 

## Figures and Tables

**Figure 1 biomolecules-13-01490-f001:**
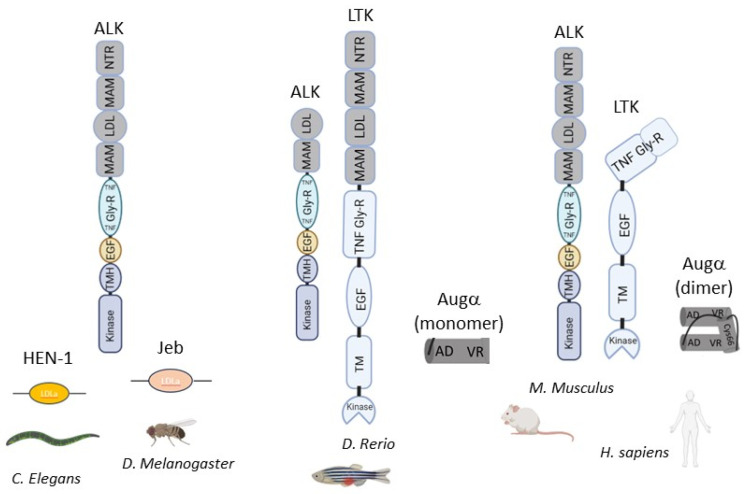
ALK and LTK receptors and species-specific respective ligands representative through evolution.

**Figure 2 biomolecules-13-01490-f002:**
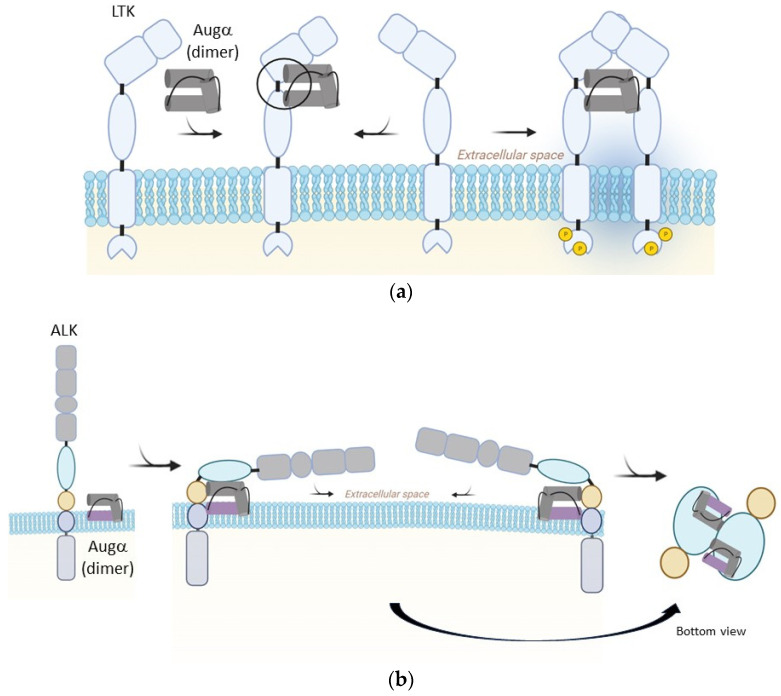
Receptor–ligand interactions. (**a**) LTK and Augα proposed mechanism of interaction by Savvides research group; (**b**) ALK and Augα proposed mechanism of interaction by Schlessinger, Klein, and Kalodimos research groups.

**Figure 3 biomolecules-13-01490-f003:**
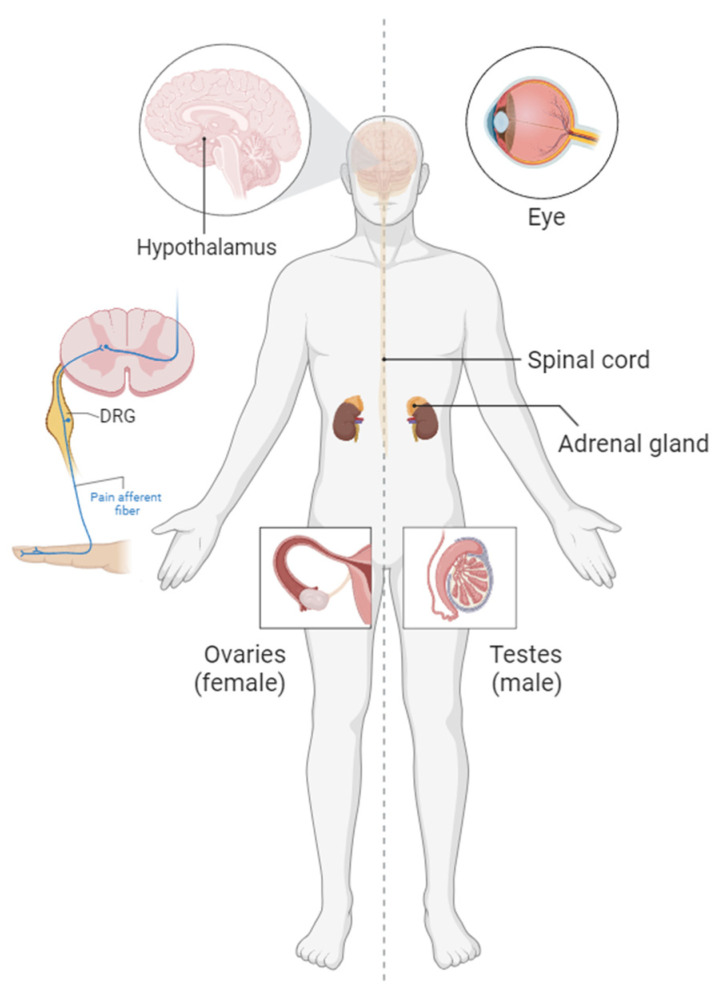
Organ-specific roles of ALK and Augα signaling axis.

**Table 1 biomolecules-13-01490-t001:** Comparison of structural studies on ALK family receptors and Augα.

Research Group	Truncations		EGF-like Domain Importance	Size Exclusion Chromatography (Receptor:Ligand)	Method Used	Structural Complex (Receptor:Ligand)
	ALK	Augα				
Savvides	TG-EGFL (648–1038)	AD (78–152)	Yes	No shift (1:1)	X-ray crystallography	1:2
Klein, Schlessinger	TG-EGFL (678–1030)	AD (71–152)	Yes	(fusion protein)	X-ray crystallography	2:2
Kalodimos, Schlessinger	TG-EGFL (673–1025)	Full size + C66Y	Yes	Shift (2:2)	cryoEM NMR	2:2

## Data Availability

No new data was created.

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
