# Peer review of "Multifaceted Roles of ALK Family Receptors and Augmentor Ligands in Health and Disease: A Comprehensive Review"

_biomolecules, 2023, doi:10.3390/biom13101490_

Round 1

Reviewer 1 Report

1. Illustrations should be inserted explaining the paragraphs (lines 64~88, 180~238).

2. It will be good to explain briefly ALK inhibitors used in the clinic, especially characteristics of brain penetration.

3. The names of drugs should be written consistently. lorlatinib vs Loratinib

4. typo: line 308 neuropahty, line 434 Trka -> TrkA, line 346 PDGFRb

Author Response

Response to Reviewer Comments

1. Reviewer Comment: ". Illustrations should be inserted explaining the paragraphs (lines 64~88, 180~238)."

Author's Response 1: Three new illustrations have been added to address the reviewer's request. These illustrations encompass an evolutionary phylogenetic tree illustrating relationships between ALK, LTK, and Aug alpha, a depiction of proposed mechanisms underlying ALK-Aug alpha and LTK-Aug alpha interactions at the cell surface, and a graphical representation highlighting the physiological functions of ALK and Aug alpha in humans. However, it's important to note that we intentionally omitted cancer-related illustrations to align with our focus on physiological roles rather than cancer, which is extensively covered in the existing literature. 

2. Reviewer Comment: "It will be good to explain briefly ALK inhibitors used in the clinic, especially characteristics of brain penetration."

Author's Response 2: While we appreciate the suggestion, elaborating extensively on FDA-approved drugs inhibiting ALK and other ALK-related drugs in preclinical and clinical trials might diverge from the primary scope we aimed to achieve, which is to emphasize the review's focus on the physiologic function of this axis. Therefore, we have chosen not to include extensive details on clinical ALK inhibitors in this manuscript.

3. Reviewer Comment: "The names of drugs should be written consistently. lorlatinib vs Loratinib."

Author's Response 3: Noted, and we have made the necessary corrections for consistency in drug names.

4. Reviewer Comment: "Typo: line 308 neuropahty, line 434 Trka -> TrkA, line 346 PDGFRb."

Author's Response 4: We appreciate your attention to detail and have corrected the noted typographical errors in the manuscript.

Reviewer 2 Report

The submitted review manuscript, “Multifaceted Roles of ALK Family Receptors and ALKAL Ligands in Health and Disease: A Comprehensive Review,” by Luka Katic and Anamarija Priscan summarizes the last 10 years of research on ligand/receptor discovery, biophysical and cellular expression, as well as known involvement in cellular functions in different model systems, including human diseases. The manuscript is of high interest to readers and objectively summarizes knowledge in the field. It is also timely submitted as the first comprehensive review dealing with the ALKAL ligands and the ALK receptor.

Specific points:

1. In the review, the authors should change the name on all instances to ALKAL1 and ALKAL2, previously named FAM150 A and B/Aug alpha or beta and then have FAM150 and AUG in brackets. This will make it easier to read, which is an important aspect. The name ALKAL was suggested by HUGO Gene Nomenclature Committee years ago, and it was agreed upon by all the authors in the USA and Europe that discovered ALKALs and are working in the field.

2. To make this review more interesting for the reader and easier to read, the authors should include one or two figures, such as a figure of the different human/mouse, drosophila, zebrafish, and c. elegans receptors/ligand complexes. A figure explaining the structure of the ALK receptor with the new interaction’s domains with ALKAL, and an overview figure of ALK/ALKALs importance for cellular function in man and mouse would also be useful.

3. Author affiliation: There are two number 2; change one to 3.

4. A point of knowledge: The authors mention that there are approximately 58 Receptro Tyrosine Kinases. However, three of them are actually Ser/The receptor kinases, which are not important to include.

5. Line 37: Missing an abbreviation for receptor tyrosine kinase (RTK).

6. Lines 135 and 136: The sentence "However, some inconsistencies were observed…" is subjective and unnecessary. It is sufficient to present the conclusion and views from the 2:1 researcher and then present the conclusion and views from the 2:2 researchers, as this is an unsolved matter, as mentioned on line 156 and onwards.

7. Lines 164 onwards: The authors should explain with a few sentences that ALK fusion proteins do not involve the ALKAL ligands. The authors could even have a small figure with Wild type ligand receptor/ligand, ALK receptors with a constitutive active kinase domain, and ALK fusion proteins.

8. Line 185: Downstream signaling proteins, such as RAS, should be changed to RAS/MAPK pathway, which includes NF1 mutations and RAF mutations.

OK

Author Response

Response to reviewer comments

1. Reviewer Comment: "In the review, the authors should change the name on all instances to ALKAL1 and ALKAL2, previously named FAM150 A and B/Aug alpha or beta and then have FAM150 and AUG in brackets. This will make it easier to read, which is an important aspect. The name ALKAL was suggested by HUGO Gene Nomenclature Committee years ago, and it was agreed upon by all the authors in the USA and Europe that discovered ALKALs and are working in the field."

Author's Response 1: I respectfully decline the proposed modification, as my training laboratory introduced the nomenclature "Aug alpha" and "Aug beta" for these ligands to distinguish them from other substances within the FAM and ALKAL families. I aim to preserve the legacy of my laboratory in this regard.

2. Reviewer Comment: "To make this review more interesting for the reader and easier to read, the authors should include one or two figures, such as a figure of the different human/mouse, drosophila, zebrafish, and c. elegans receptors/ligand complexes. A figure explaining the structure of the ALK receptor with the new interaction’s domains with ALKAL, and an overview figure of ALK/ALKALs importance for cellular function in man and mouse would also be useful."

Author's Response 2: Acknowledged, and as suggested, we have incorporated three graphical representations into the manuscript. These include an evolutionary phylogenetic tree depicting ALK, LTK, and Aug alpha, an illustrative depiction of the proposed mechanisms underlying the interactions between ALK and Aug alpha, and LTK and Aug beta at the cellular surface, as well as a graphical representation highlighting the physiological functions in humans.

3. Reviewer Comment: "Author affiliation: There are two number 2; change one to 3."

Author's Response 3: The typographical error has been corrected, and the author affiliation numbering now aligns correctly.

4. Reviewer Comment: "A point of knowledge: The authors mention that there are approximately 58 Receptro Tyrosine Kinases. However, three of them are actually Ser/The receptor kinases, which are not important to include."

Author's Response 4: We appreciate the clarification regarding the inclusion of receptor kinases. Our revised manuscript now accurately represents this knowledge.

5. Reviewer Comment: "Line 37: Missing an abbreviation for receptor tyrosine kinase (RTK)."

Author's Response 5: We have duly addressed the omission of the abbreviation for receptor tyrosine kinase (RTK) on line 37.

6. Reviewer Comment: "Lines 135 and 136: The sentence 'However, some inconsistencies were observed…' is subjective and unnecessary. It is sufficient to present the conclusion and views from the 2:1 researcher and then present the conclusion and views from the 2:2 researchers, as this is an unsolved matter, as mentioned on line 156 and onwards."

Author's Response 6: The manuscript has been revised to reflect the suggestion for a more balanced presentation. We acknowledge the importance of discussing potential influences of size exclusion chromatography on crystallographic conditions within this unresolved matter.

7. Reviewer Comment: "Lines 164 onwards: The authors should explain with a few sentences that ALK fusion proteins do not involve the ALKAL ligands. The authors could even have a small figure with Wild type ligand receptor/ligand, ALK receptors with a constitutive active kinase domain, and ALK fusion proteins."

Author's Response 7: The manuscript now underscores the critical point that ALK fusion proteins do not involve interactions with ALKAL ligands more explicitly and comprehensibly. The omission of cancer-related illustrations is intentional, as the review focuses primarily on physiological roles rather than cancer and FDA-approved therapeutics, which are extensively covered in the existing literature.

8. Reviewer Comment: "Line 185: Downstream signaling proteins, such as RAS, should be changed to RAS/MAPK pathway, which includes NF1 mutations and RAF mutations."

Author's Response 8: We have duly acknowledged and implemented the suggested modification, replacing "Downstream signaling proteins, such as RAS" with "RAS/MAPK pathway," encompassing NF1 mutations and RAF mutations within this context.